# Continual Learning using Evolution Strategies

## Abstract

Continual Learning (CL) aims to train neural networks on sequences of tasks without triggering catastrophic forgetting. Existing approaches typically rely on gradient-based optimization, which breaks down in exemplar-free settings where data and therefore gradients from past tasks are unavailable. To overcome this limitation, we propose EvoCL, a gradient-free method that employs an evolutionary strategy to optimize neural network using a surrogate loss constructed by an adapter network. The adapter maps stored latent features of previous classes into the current task's embedding space, enabling joint training of the feature extractor and adapter without access to past data or gradients. This reframes CL as an optimization problem that does not require gradient information. Experiments on multiple benchmarks demonstrate that EvoCL achieves strong performance under tight parameter budgets, highlighting it as a promising direction for gradient-free, exemplar-free CL. The code to reproduce these results is available at (omitted for the review).

## 1 Introduction

Continual learning (CL) presents a fundamental challenge in training neural networks on sequential tasks without experiencing catastrophic forgetting French (1999). Most existing CL approaches rely on backpropagation-based optimization, where model parameters are updated via stochastic gradient descent (SGD) or its variants Masana et al. (2022); Zhou et al. (2023). However, in exemplar-free settings Li & Hoiem (2017); Petit et al. (2023); Goswami et al. (2024); Rypeść et al. (2023); Zhuang et al. (2022) -where data from previous tasks cannot be stored due to limited memory or privacy concerns - gradients from past tasks are unavailable. This leads to uncontrolled parameter drift Yu et al. (2020) and rapid forgetting, especially when tasks are disjoint.

Several prior methods attempt to indirectly mitigate this issue by regularizing parameter updates Kirkpatrick et al. (2017); Zenke et al. (2017); Farajtabar et al. (2020) or distilling knowledge from previous models Li & Hoiem (2017); Yu et al. (2020); Magistri et al. (2024). While effective to some extent, these techniques still depend on gradient flow through the model, limiting their ability to optimize for past tasks once their gradients are gone.

To address this limitation, we propose reframing CL as an optimization problem under gradient unavailability. We introduce EvoCL, an evolutionary-based approach to Exemplar-Free Class-Incremental Learning (EFCIL) that decouples the optimization process from the necessity of backpropagation. By leveraging a population-based $(\mu + \lambda)$ Evolution Strategy (ES) Rechenberg (1970); Slowik & Kwasnicka (2020), EvoCL can directly minimize an objective function that includes an approximated loss for past tasks,

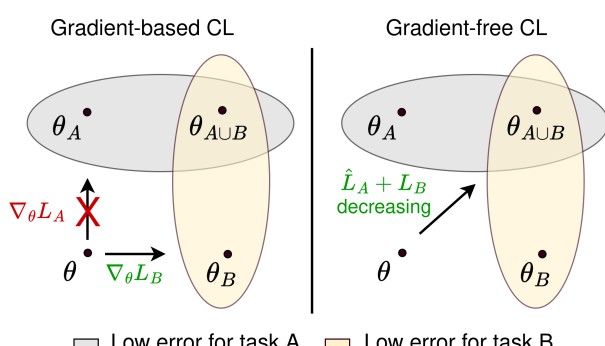

Figure 1: Gradient-based optimizers fail to find a joint minimum suitable for both tasks A and B in continual learning due to the lack of gradients from task A. However, given a non-differentiable approximation of task A's loss $L_A$ (denoted $\hat{L}_A$), a gradient-free optimizer can identify a satisfactory solution.

even when the gradients for those tasks are non-existent or non-differentiable - as depicted in Fig. 1. To facilitate this approximation without storing raw data, we maintain a compact buffer of latent features and utilize an auxiliary adapter network that dynamically transforms past embeddings into the current latent space. This architecture allows EvoCL to perform cross-task optimization through reproduction, mutation, and selection, effectively combating catastrophic forgetting by exploring the parameter space for solutions that satisfy both current and historical task constraints. Our method not only maintains memory efficiency by avoiding covariance matrix storage but also demonstrates that gradient-free optimization is a robust alternative for navigating the rigid constraints of exemplar-free environments. Our contributions can be summarized as follows:

- We shift the focus in **the fundamental research of continual learning** from overcoming the lack of past data to overcoming the lack of past gradients.

- We propose EvoCL, a novel gradient-free method that replaces backpropagation with an evolution strategy for subsequent tasks, enabling effective optimization even when past-task gradients are unavailable.

- We present the first exemplar-free continual learning approach that jointly trains both the feature extractor and a feature adaptation network.

## 2 Related works

**Class-Incremental Learning (CIL)** is widely regarded as one of the most difficult and commonly studied settings within Continual Learning Van de Ven & Tolias (2019); Masana et al. (2022). In this scenario, the model must predict over all classes encountered so far without access to task identifiers during inference. A straightforward approach to mitigate catastrophic forgetting in CIL involves storing exemplar samples, as seen in methods like LUCIR Hou et al. (2019), BiC Wu et al. (2019), Foster Wang et al. (2022), and WA Zhao et al. (2020). These stored examples help facilitate the learning of cross-task representations. However, exemplar storage is often impractical due to privacy concerns or storage constraints. This has led to the development of exemplar-free CIL (EFCIL) methods, including LwF Li & Hoiem (2017), SDC Yu et al. (2020), ABD Smith et al. (2021), PASS Zhu et al. (2021b), IL2A Zhu et al. (2021a), SSRE Zhu et al. (2022), and SEED Rypeść et al. (2024b). These approaches typically emphasize stability and combat forgetting by applying regularization techniques to a strong feature extractor. Some, such as FeTrIL Petit et al. (2023) and FeCAM Goswami et al. (2024), even restrict learning to the classifier, keeping the feature backbone fixed. While this can preserve stability, it may hinder adaptability - especially in more complex or heterogeneous task sequences, as observed in works like CTrL Veniat et al. (2020).

**Gradient-Free Optimization** methods have gained attention as alternatives to gradient-based techniques for training neural networks, especially in scenarios where gradients are unavailable, unreliable, or expensive to compute. These methods include evolutionary algorithms, Bayesian optimization, and zeroth-order optimization, which optimize model parameters by evaluating objective functions directly rather than relying on gradient information. Evolutionary strategies, such as the Covariance Matrix Adaptation evolution strategy - CMA-ES Hansen & Ostermeier (2001), have been successfully applied to small neural network training, demonstrating strong performance in black-box settings Hansen et al. (2003); Salimans et al. (2017). Similarly, Bayesian optimization has proven effective for hyperparameter tuning and small-scale neural networks, balancing exploration and exploitation in parameter search Snoek et al. (2012). Zeroth-order methods, such as those explored in Nesterov & Spokoiny (2017) and Liu et al. (2020), estimate gradients via finite differences or random sampling, allowing optimization in non-differentiable or adversarial environments. More recent approaches like Direct Feedback Alignment Nøkland (2016), evolutionary gradient methods Such et al. (2017) and Forward Forward algorithm Hinton (2022) have shown that neural networks can be trained competitively without backpropagation, challenging the necessity of gradient flow in deep learning. While gradient-free methods are generally less efficient than backpropagation in standard settings, they offer robustness and applicability in constrained or novel problem domains where gradient access is limited or costly.

**Task-to-Task feature transformation** Regularization-based approaches in EFCIL penalize changes to important neural network parameters Kirkpatrick et al. (2017); Chaudhry et al. (2018); Zenke et al. (2017);

---

**Algorithm 1** *EvoCL*: Evolutionary Continual Learning pseudocode

---

1: **Initialize:** Training data $(D_1, D_2, \ldots, D_T)$, feature extractor $F_1$, $\alpha$, empty buffer $M_0$ for memorized features
2: Train $F_1$ on $D_1$ using SGD optimizer and CE loss
3: **for** class $c \in C_1$ **do**
4:     Add class features into the buffer $M_1 = M_0 \cup \{F_1(x) : x, c \in C_1\}$
5: **end for**
6: **for** task $t = 2, 3, \ldots, T$ **do**
7:     Initialize $\psi^{t-1 \to t}$ (auxiliary adapter network - MLP) as an identity function
8:     Train $F_t$ and $\psi^{t-1 \to t}$ on $D_t \cup \psi^{t-1 \to t}(M_{t-1})$ using $L_{EvoCL} = L_t + \widehat{L}_{<t} + \alpha \cdot L_{\mathrm{MSE}}$ and ES optimizer
9:     Adapt features: $M_t = \{\psi^{t-1 \to t}(m) : m \in M_{t-1}\}$
10:     **for** class $c \in C_t$ **do**
11:         Add class features into the buffer $M_t = M_{t-1} \cup \{F_t(x) : x, c \in D_t\}$
12:     **end for**
13: **end for**

---

Liu et al. (2018) or use distillation techniques to regularize neuron activations Li & Hoiem (2017); Yu et al. (2020); Zhu et al. (2021a); Magistri et al. (2024). However, even with knowledge distillation, the features of old classes change from task to task, causing catastrophic forgetting French (1999); McCloskey & Cohen (1989). Therefore, few works tried to predict these changes by approximating their semantic drift Yu et al. (2020); Magistri et al. (2024); Iscen et al. (2020); Rypeść et al. (2024b) and applying a correction after each task. However, those strategies' limitations are that they adapt features, ignoring changes in covariance matrices, which is suboptimal. Therefore, Rypeść et al. (2024a) proposed to also adapt the variance of features distribution. In this work we utilize the idea of the feature adaptation, however we train the adapter network jointly with the feature extractor to improve separability of classes.

## 3 Method

In this section we describe motivation and EvoCL - our CL method that leverages gradient-free optimization and loss approximation via feature transformation to combat catastrophic forgetting. EvoCL is designed for the Exemplar-Free Class-Incremental Learning (EFCIL) setting and relies on evolution strategies to optimize neural networks without access to data from previous tasks. The pseudocode of the method is presented in Alg. 1 and the overview of it is depicted in Fig. 2.

**Exemplar-Free Class-Incremental Learning (EFCIL)** In EFCIL, the dataset is partitioned into $T$ sequential tasks $D_1, D_2, ..., D_T$, each composed of unique, non-overlapping class sets $C_t$. At any step $t$, only current task data $D_t$ is available; no previous task data can be stored or revisited. The goal is to train a classifier $F_t(\theta)$, parametrized with a vector of weights $\theta$, that can discriminate among all classes $\bigcup_{i=1}^{t} C_i$ encountered so far, without any information about the current task ID at inference (i.e., task-agnostic evaluation Masana et al. (2022)).

**Theoritical motivation** Let $\theta^t$ denote the parameters after task $t$ and $L_t(\theta)$ the task-specific cross-entropy loss. In CL, the true objective at step $t$ is:

$$L_{\leq t}(\theta) = \sum_{i=1}^{t} L_i(\theta^t). \tag{1}$$

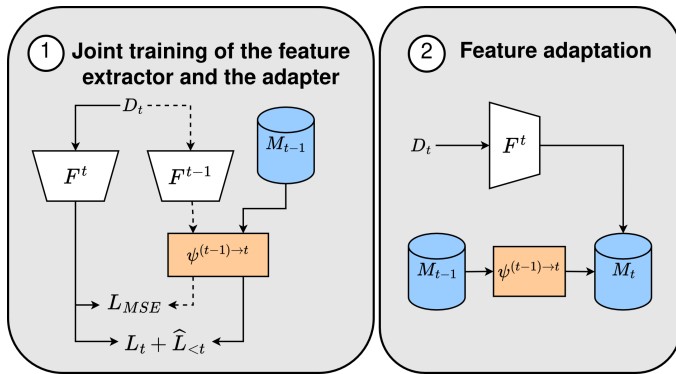

Figure 2: Each task involves a two-step process: (1) Joint training, where the feature extractor $F^t$ and adaptation network $\psi^{(t-1) \to t}$ are optimized (dashed lines denote data flow through the frozen $F^{t-1}$); and (2) Feature adaptation, where previously memorized features are mapped into the current latent space and integrated into the features buffer $M$ together with new features from the current task.

However, because past data is unavailable, $\nabla_\theta L_{<t}(\theta^t)$ cannot be directly computed. This makes the update biased toward only the current task, i.e. $\nabla_\theta L_t(\theta^t) \approx \nabla_\theta L_{\leq t}(\theta^t)$ no longer holds, which leads to forgetting.

Conversely, gradient-free methods such as ES optimize parameters by direct function evaluations:

$$\theta^{t+1} = \arg\min_{\theta \in \mathcal{P}_t} L_{1:t}(\theta),$$

where $\mathcal{P}_t$ is a population of perturbed candidates. They do not require access to $\nabla_\theta L_{<t}$ and thus can minimize surrogate approximations $\widehat{L}_{<t} : \forall_{i \leq t} \widehat{L}_i$ without gradients.

The idea behind EvoCL is to represent Eq. 1 as the sum of surrogate losses (assuming $t > 1$; for $t = 1$ we use SGD) and minimize it using ES optimizer. Therefore, the fitness function is equal to:

$$L_{\leq t}(\theta) = \sum_{i=1}^t L_i(\theta^t) \approx \sum_{i=1}^{t-1} \widehat{L}_i(\theta^t) + L_t(\theta)$$

for any task $t > 1$. In order to compute $\widehat{L}_{<t}$ we utilize a buffer of latent features from tasks $< t$ and an adaptation network to transform these features to the current task's latent space, similarly to Iscen et al. (2020). This decouples continual optimization from gradient availability.

**Latent Buffer and Feature Memorization** After training on task $t$, we extract and store a small number of $N$ latent features per class from the trained feature extractor $F_t$. These features are stored in a memory buffer $M$ and represent each class in a compressed, forward-pass-only form. Unlike traditional rehearsal-based methods, EvoCL does not store original input samples - only intermediate embeddings. This design keeps memory usage low and complies with the privacy and exemplar-free constraints, while retaining task-relevant information that can be reused via adapter transformations.

**Adapter Network for Loss Approximation** The adapter network $\psi^{t-1 \to t} : S \to S$, where $S$ is the size of the latent space, is an auxiliary neural module (e.g., MLP) trained to map features from the latent space of $F_{t-1}$ to that of $F_t$. During task $t$, the adapter takes features $m \in M$ stored in past tasks $< t$ and transforms them into the embedding space of the current feature extractor. This transformation allows computing an approximated cross-entropy loss $\widehat{L}_{<t}$ using only the memorized embeddings from buffer $M$.

To train the adapter, we minimize a mean squared error loss $L_{MSE}$ between the transformed features $\psi^{t-1 \to t}(F_{t-1}(x))$ and $F_t(x)$, where $x$ is sampled from the current task data $D_t$. The adapter is discarded after training on a task, thus is does not increase the memory requirement and computational complexity of the method. The adapter's weights are initialized so that the adapter is an identity function, because at the beginning of the training $F_t = F_{t-1}$ (e.g. if the adapter is a linear layer, its weight matrix is set to identity matrix and biases are equal to 0).

**Gradient-Free Optimization via Evolution Strategy** The loss approximated for past tasks $(\widehat{L}_{<t})$ is a result of transforming memorized features from the latent buffer through the adapter:

$$\widehat{L}_{<t}(\theta_t, \psi) = \sum_{(m,y) \in B_M} \text{CE}\big( f_t^{\text{cls}}\big(\psi^{t-1 \to t}(m)\big), y\big)$$

where $B_M \subseteq M$ is a mini-batch of stored latent features $m \in \mathbb{R}^S$ with corresponding labels $y \in \bigcup_{i=1}^{t-1} C_i$, $f_t^{\text{cls}}$ is the classifier head of the current feature extractor $F_t$, and $\text{CE}(\cdot, \cdot)$ denotes the cross-entropy loss. Therefore, the gradient $\nabla_\theta \widehat{L}_{<t}$ cannot be computed and gradient-based optimizers will fail to optimize $F_t$. For this reason, to optimize feature extractor $F_t$ with respect to $\widehat{L}_{<t}$ we use a popular $(\mu + \lambda)$ evolutionary strategy Rechenberg (1970). This method enables direct parameter optimization through population-based search, making it suitable for scenarios with non-differentiable components like our adapter-based loss approximation. Following standard CL protocols, the first task is optimized via standard SGD to establish a robust initial latent space, whereas all subsequent task transitions are optimized using our gradient-free evolution strategy.

In our ES optimizer, we maintain a population of $\mu$ parent models. Each such model $i < \mu$ (called individual) is a pair of $\{\theta_i, \psi_i^{t-1 \to t}\}$. Training process is split into epochs which consist of iterations. In each iteration a

mini-batch of data is presented to the population. When the population is presented all the mini-batches of data $D_t$ the next epoch begins. All individuals share the same latent feature buffer $M$. At each iteration we perform the following steps:

- **Reproduction (crossover):** $\lambda$ offsprings are created via interpolation-based crossover, where for each offspring we sample with replacement parents $i$ and $j$:

$$\theta_{\text{child}} = \beta \cdot \theta_i + (1 - \beta) \cdot \theta_j,$$
$$\psi_{\text{child}}^{t-1 \to t} = \beta \cdot \psi_i^{t-1 \to t} + (1 - \beta) \cdot \psi_j^{t-1 \to t}$$
$$\text{where} \quad \beta \sim \mathcal{U}(0, 1) \quad \text{and} \quad 0 \le i, j < \mu$$

- **Mutation:** Gaussian noise is added to the offspring to introduce variation:

$$\theta_{\text{child}} \leftarrow \theta_{\text{child}} + \epsilon_\theta, \quad \epsilon_\theta \sim \mathcal{N}(0, \sigma^2 I),$$
$$\psi_{\text{child}}^{t-1 \to t} \leftarrow \psi_{\text{child}}^{t-1 \to t} + \epsilon_\psi, \quad \epsilon_\psi \sim \mathcal{N}(0, \sigma^2 I),$$
$$\text{where } \sigma \text{ is the mutation strength hyperparameter.}$$

- **Evaluation:** Each individual (parent and offspring) is scored using the combined loss function:

$$L_{EvoCL} = L_t + \widehat{L}_{<t} + \alpha \cdot L_{\text{MSE}} \tag{2}$$

  Here, $\alpha$ balances classification performance and adapter reconstruction quality. It is a hyperparameter common for all individuals.

- **Selection:** The top $\mu$ individuals with the lowest loss values $L_{EvoCL}$ are retained for the next iteration.

After the final epoch, the best individual from the parent set is chosen as the model (according to $L_{EvoCL}$ on validation set), while other individuals are discarded.

This approach allows both exploitation of high-performing solutions and exploration of new ones. The interpolation-based crossover ensures smooth inheritance, while Gaussian mutation encourages diversity, making this strategy effective in gradient-inaccessible EFCIL settings. Compared to modern evolution strategies based on covariance matrices, e.g. Hansen & Ostermeier (2001); Wierstra et al. (2014); Maheswaranathan et al. (2019), our approach reduces memory and computational complexity what is crucial for training sizeable neural networks in parallel on GPU (>10k parameters).

**Task-to-Task Feature Transformation** After training on task $t$, we forward all stored features in buffer $M$ through the adapter $\psi^{t-1 \to t}$ to align them with the new latent space. This cumulative transformation ensures that the buffer remains valid across all subsequent tasks. This step is presented in the ninth state of Alg. 1. This transformation was introduced in Iscen et al. (2020) and it aligns all prior class representations to the current model's embedding space, enabling continual reuse in approximating $L_{<t}$ at future steps.

**Memory complexity** EvoCL requires $\theta + |M| \cdot S$ parameters to run, where $S$ is the latent space size. This makes it on par with methods that also store embeddings in the memory buffer, such as Iscen et al. (2020). EvoCL requires less parameters to run than methods that memorize covariance matrices per class Goswami et al. (2024); Rypeść et al. (2023; 2024a), provided that $N < \frac{S-1}{2}$. This requirement is fulfilled in our experiments ($N = 64, S = 192$).

## 4 Experiments

**Datasets and metrics.** We conduct experiments on five well-established EFCIL benchmark datasets, evaluating both training from scratch and fine-tuning from a pre-trained Vision Transformer Dosovitskiy et al. (2021). Our benchmark includes simple datasets such as MNIST Deng (2012) and FashionMNIST Xiao et al.

(2017), each comprising 60,000 training and 10,000 test images across 10 classes. For a more challenging setup, we include CIFAR100 Krizhevsky (2009), which contains 50,000 training and 10,000 test images distributed over 100 classes. We further evaluate our method on FGVCAircraft Maji et al. (2013), a fine-grained classification dataset of 10,200 airplane images belonging to 100 classes. Additionally, to assess robustness under distributional shifts, we conduct experiments on DomainNet Peng et al. (2019), which contains classes across six distinct domains. For this benchmark, each task is constructed from a subset of classes within a single domain, ensuring that the domain changes across tasks - thereby introducing significant domain drift as in Rypeść et al. (2023).

To simulate the incremental learning scenario, we partition each dataset into $T$ tasks, where each task contains a disjoint subset of classes. We do not use exemplars and assume that task id is unknown during test time. This setup follows standard EFCIL protocols Li & Hoiem (2017); Yu et al. (2020); Rypeść et al. (2023); Magistri et al. (2024).

For the evaluation metric, we utilize commonly used average accuracy $A_{last}$, which is the accuracy after the last task, and average incremental accuracy $A_{inc}$, which is the average of accuracies after each task Masana et al. (2022); Petit et al. (2023); Goswami et al. (2024). We repeat experiments three times and report mean and standard deviations of the metrics.

**Baselines and hyperparameters.** We compare EvoCL method to multiple EFCIL baselines. Well-established ones, like LwF Li & Hoiem (2017), EWC Kirkpatrick et al. (2017), PASS Zhu et al. (2021c), IL2A Zhu et al. (2021a), and the most recent and strong EFCIL baselines: FeTrIL Petit et al. (2023), FeCAM Goswami et al. (2024) and DS-AL Zhuang et al. (2024). We also provide results for naive finetuning of the neural network dubbed *Finetune* which has no means to combat forgetting. Our *Upper bound* is the model trained on all accumulated task datasets simultaneously. This serves as the performance ceiling where catastrophic forgetting is absent. We run implementations provided in FACIL Masana et al. (2022) and PyCIL Zhou et al. (2021) frameworks (if provided) or from the authors' repositories. For each method we set hyperparameters to default as proposed in the original works. We utilize random crops and horizontal flips as data augmentation. Details can be found in the *scripts* directory of the Supplementary Material.

**Implementation details and reproducibility.** For every method we utilize the same neural network as the feature extractor $F$. For MNIST, Fashion and CIFAR100 datasets we utilize a 4-layer convolutional network containing 20k parameters and the latent space size equal to 32. For DomainNet and FGVCAircraft datasets we utilize ViT-small Dosovitskiy et al. (2021) model pretrained with DINO Caron et al. (2021) on ImageNet Deng et al. (2009). We add a small MLP network before the MLP layer in the last block that contains 30k trainable parameters and keep the rest frozen.

We implement our method in FACILMasana et al. (2022) benchmark. We implement the ES optimizer using PyTorch library Paszke et al. (2017) so that it can run on GPU and all computations regarding individuals are parallelized for faster training. We set $\alpha = 100, \mu = 16, \lambda = 128$ and starting mutation strength $\sigma$ to 1e-4. We linearly decrease $\sigma$ to $1e-5$ during training. As the adapter network $\psi^{t-1 \rightarrow t}$ we utilize a two layer MLP network with hidden size equal to 16. On MNIST and FashionMNIST we train EvoCL for 200 epochs, while on other - more challenging datasets we train for 2000 epochs to ensure convergence.

We utilize a single machine with an NVIDIA RTX4080 graphics card to run experiments. The time for execution of a single experiment varied depending on the dataset type, but it was at most twenty hours. We attach details of utilized hyperparameters in scripts in the code repository. We report all results as the mean and variance of five runs.

**Results when training from scratch** are provided in Tab. 1. The EvoCL method demonstrates strong and consistent performance across all evaluated datasets and task configurations, outperforming all competing baselines in both average incremental accuracy ($A_{inc}$) and final task accuracy ($A_{last}$). On MNIST, EvoCL achieves the highest $A_{last}$ scores of 92.3% (T=3) and 81.8% (T=5), significantly surpassing strong baselines such as DS-AL and FeCAM. Its $A_{inc}$ values, 95.8% and 89.9% for T=3 and T=5 respectively, indicate that EvoCL maintains robust performance throughout the continual learning process, staying remarkably close to the offline upper bound of 98.7%.

Table 1: Average incremental and last accuracy in EFCIL for different datasets and number of tasks $T$ when training from scratch. We report the mean and standard deviation of three runs. Gradient-free approach (EvoCL) yields very promising results.

| Method | MNIST | | | | FashionMNIST | | | | CIFAR100 | | | |
|---|---|---|---|---|---|---|---|---|---|---|---|---|
| | $T$=3 | | $T$=5 | | $T$=3 | | $T$=5 | | $T$=5 | | $T$=10 | |
| | $A_{last}\uparrow$ | $A_{inc}\uparrow$ | $A_{last}\uparrow$ | $A_{inc}\uparrow$ | $A_{last}\uparrow$ | $A_{inc}\uparrow$ | $A_{last}\uparrow$ | $A_{inc}\uparrow$ | $A_{last}\uparrow$ | $A_{inc}\uparrow$ | $A_{last}\uparrow$ | $A_{inc}\uparrow$ |
| Upper bound | 98.7±0.1 | | | | 92.1±0.1 | | | | 71.8±0.2 | | | |
| Finetune | 40.2±0.2 | 65.9±0.2 | 21.6±0.2 | 52.8±0.2 | 21.1±0.2 | 54.8±0.2 | 27.9±0.2 | 40.2±0.2 | 14.7±0.2 | 17.1±0.2 | 12.6±0.1 | 14.0±0.2 |
| LwF | 85.6±0.1 | 91.3±0.1 | 47.4±0.2 | 72.6±0.2 | 27.0±0.1 | 57.9±0.2 | 32.1±0.1 | 50.9±0.1 | 22.1±0.2 | 34.1±0.2 | 19.1±0.1 | 31.1±0.1 |
| EWC | 76.7±0.2 | 82.5±0.1 | 44.8±0.2 | 69.2±0.2 | 26.5±0.2 | 54.3±0.2 | 30.0±0.2 | 48.8±0.1 | 20.9±0.1 | 30.3±0.1 | 17.4±0.2 | 28.6±0.2 |
| PASS | 53.7±0.3 | 69.8±0.2 | 44.8±0.2 | 55.7±0.2 | 22.7±0.1 | 56.9±0.1 | 28.5±0.3 | 42.8±0.2 | 18.1±0.1 | 29.0±0.1 | 17.3±0.3 | 30.9±0.2 |
| IL2A | 56.9±0.1 | 61.3±0.1 | 29.4±0.1 | 45.5±0.1 | 20.8±0.1 | 35.1±0.1 | 28.4±0.1 | 43.1±0.1 | 19.3±0.1 | 31.1±0.1 | 14.2±0.1 | 28.7±0.1 |
| FeTrIL | 78.3±0.1 | 80.9±0.1 | 75.5±0.2 | 79.9±0.2 | 61.6±0.2 | 68.1±0.1 | 59.5±0.2 | 62.0±0.2 | 21.7±0.1 | 34.1±0.2 | 18.8±0.1 | 30.5±0.1 |
| FeCAM | 79.4±0.1 | 82.5±0.1 | 76.2±0.1 | 81.4±0.1 | 64.0±0.3 | 70.3±0.2 | 63.8±0.1 | 67.1±0.2 | 22.2±0.1 | 34.7±0.1 | 20.6±0.2 | 32.9±0.2 |
| DS-AL | 84.2±0.2 | 87.8±0.2 | 75.4±0.2 | 77.0±0.2 | 73.6±0.1 | 75.3±0.2 | 68.1±0.2 | 64.1±0.2 | 21.7±0.1 | 35.6±0.2 | 20.0±0.2 | 32.6±0.2 |
| **EvoCL** | **92.3**±0.3 | **95.8**±0.2 | **81.8**±0.2 | **89.9**±0.2 | **77.2**±0.4 | **78.0**±0.2 | **72.0**±0.2 | **74.9**±0.2 | **24.8**±0.2 | **37.2**±0.1 | **21.3**±0.1 | **35.8**±0.1 |

On the more challenging FashionMNIST, EvoCL again leads, reaching 77.2% and 72.0% $A_{last}$ for T=3 and T=5 respectively. These results outperform the closest competitors (e.g., DS-AL and FeCAM) by a clear margin of 3–5 percentage points. Notably, EvoCL achieves an $A_{inc}$ of 78.0% (T=3) and 74.9% (T=5), showing its ability to retain previously learned knowledge effectively while learning new tasks.

In the case of CIFAR100, which poses a more complex and high-variance setting, EvoCL shows modest but still leading gains over existing methods. It achieves 24.8% (T=5) and 21.3% (T=10) in $A_{last}$, along with 37.2% and 35.8% in $A_{inc}$, outperforming the next-best baselines by up to 2 percentage points. While the absolute accuracies are lower due to dataset difficulty, the relative improvements highlight EvoCL's capacity to scale to more challenging scenarios.

**Results when finetuning a pre-trained model.** Tab. 2 presents a comprehensive comparison of EvoCL against several state-of-the-art baselines in EFCIL fine-grained scenarios using a pre-trained feature extractor. The results clearly show that EvoCL consistently outperforms all competing methods across various task configurations on both the DomainNet and FGVCAircraft datasets.

On DomainNet, EvoCL achieves the highest average last accuracy $A_{last}$ and average incremental accuracy $A_{inc}$ across all task settings. For instance, with $T = 6$, EvoCL reaches $A_{last} = 73.1\%$ and $A_{inc} = 81.6$, outperforming the second-best method, DS-AL, which achieves 70.4% and 77.9%, respectively. As the number of tasks increases, EvoCL maintains its performance advantage. At $T = 18$, it achieves $A_{\text{last}} = 43.8\%$ and $A_{inc} = 57.2\%$, compared to DS-AL's 41.4% and 53.8%. This proves solid adaptability and robustness of EvoCL across different data domains.

On FGVCAircraft, a similar trend is observed. EvoCL consistently outperforms all baselines. At $T = 5$, EvoCL records $A_{last} = 34.8\%$ and $A_{inc} = 48.2\%$, while DS-AL reaches 32.6% and 46.7%. Even at the more challenging $T = 20$ setting, EvoCL remains ahead with $A_{last} = 32.1\%$ and $A_{inc} = 52.7\%$, compared to DS-AL's 31.2% and 48.7%.

These results highlight EvoCL's ability to effectively balance stability and plasticity, enabling it to retain prior knowledge while integrating new information efficiently. Its consistent superiority across datasets and task counts demonstrates its scalability and robustness in fine-grained class-incremental learning scenarios.

**Does Evolution Strategy outperform SGD?** To investigate whether evolution strategy (ES) offers an advantage over stochastic gradient descent (SGD) in the context of our EvoCL framework, we conduct a comparative study on FashionMNIST dataset split into five tasks by replacing ES with SGD as the optimizer. To ensure a fair baseline comparison and mitigate parameter selection bias, we conducted extensive hyperparameter grids for both methods. For the SGD baseline, we swept learning rates $lr \in 0.001, 0.01, 0.1, 1.0$ and momentum terms $\mu \in 0.0, 0.5, 0.8, 0.9$, testing each combination across balancing coefficients $\alpha \in 1, 10, 100, 1000$. For SGD we utilize the same loss function (Eq. 2), as for ES.

Even when comparing the best-performing, extensively tuned SGD configuration ($lr = 0.1, \mu = 0.9, \alpha = 100$) against our default ES setup, the results, shown in Fig. 3, highlight a clear advantage of ES. In the left

Table 2: Average incremental and last accuracy in EFCIL fine-grained scenarios when utilizing a pre-trained feature extractor. We report the mean and standard deviation of three runs. EvoCL performs better than counterparts.

| Method | DomainNet | | | | | | FGVCAircraft | | | | | |
| | $T{=}6$ | | $T{=}12$ | | $T{=}18$ | | $T{=}5$ | | $T{=}10$ | | $T{=}20$ | |
| | $A_{last}\uparrow$ | $A_{inc}\uparrow$ | $A_{last}\uparrow$ | $A_{inc}\uparrow$ | $A_{last}\uparrow$ | $A_{inc}\uparrow$ | $A_{last}\uparrow$ | $A_{inc}\uparrow$ | $A_{last}\uparrow$ | $A_{inc}\uparrow$ | $A_{last}\uparrow$ | $A_{inc}\uparrow$ |
|---|---|---|---|---|---|---|---|---|---|---|---|---|
| Upper bound | 82.3±0.1 | | | | | | 87.0±0.1 | | | | | |
| Finetune | 21.6±0.3 | 38.2±0.2 | 15.8±0.2 | 32.6±0.2 | 12.3±0.1 | 27.2±0.2 | 24.3±0.1 | 44.0±0.1 | 14.3±0.1 | 34.5±0.2 | 10.9±0.2 | 27.9±0.1 |
| LwF | 54.3±0.1 | 67.7±0.1 | 40.4±0.1 | 54.1±0.1 | 37.4±0.1 | 45.7±0.1 | 30.0±0.1 | 44.2±0.1 | 28.0±0.2 | 46.5±0.1 | 14.7±0.1 | 30.5±0.1 |
| EWC | 21.6±0.1 | 38.2±0.2 | 15.8±0.1 | 32.6±0.1 | 12.3±0.1 | 27.2±0.1 | 24.3±0.1 | 44.0±0.1 | 14.3±0.1 | 34.5±0.2 | 10.9±0.2 | 27.9±0.1 |
| PASS | 44.5±0.3 | 58.2±0.4 | 27.0±0.3 | 42.3±0.2 | 18.1±0.3 | 36.9±0.2 | 26.3±0.2 | 42.7±0.1 | 26.4±0.1 | 41.0±0.1 | 18.9±0.3 | 28.2±0.2 |
| IL2A | 46.9±0.4 | 61.3±0.3 | 29.4±0.1 | 45.5±0.2 | 20.8±0.1 | 35.1±0.1 | 28.4±0.1 | 38.1±0.1 | 25.3±0.1 | 39.1±0.1 | 18.2±0.1 | 28.7±0.1 |
| FeTrIL | 68.9±0.2 | 73.2±0.2 | 54.9±0.3 | 61.2±0.2 | 38.6±0.3 | 49.3±0.2 | 31.0±0.3 | 45.5±0.2 | 30.5±0.2 | 48.4±0.2 | 30.5±0.3 | 43.3±0.2 |
| FeCAM | 71.5±0.1 | 78.4±0.1 | 56.2±0.2 | 66.9±0.2 | 40.2±0.1 | 50.9±0.1 | 33.3±0.2 | 47.0±0.2 | 32.9±0.2 | 50.2±0.2 | 31.0±0.1 | 50.0±0.2 |
| DS-AL | 70.4±0.2 | 77.9±0.2 | 55.8±0.2 | 64.1±0.2 | 41.4±0.2 | 53.8±0.2 | 32.6±0.3 | 46.7±0.2 | 32.3±0.2 | 51.4±0.2 | 31.2±0.2 | 48.7±0.2 |
| EvoCL | **73.1**±0.4 | **81.6**±0.3 | **63.0**±0.4 | **72.3**±0.3 | **43.8**±0.3 | **57.2**±0.1 | **34.8**±0.3 | **48.2**±0.3 | **34.6**±0.3 | **52.4**±0.3 | **32.1**±0.4 | **52.7**±0.2 |

plot, we examine the evolution of the past-task loss $L_{<t}$ during training on the second task. For all tested $\alpha$ values, SGD leads to a sharp increase in $L_{<t}$ - either early in training ($\alpha = 10, 100$) or after a short delay ($\alpha = 1000$), indicating severe forgetting. In contrast, ES consistently reduces $L_{<t}$ from approximately 1.4 to 0.6 within the first 20 epochs, demonstrating its robustness in optimizing for past tasks without requiring gradient information.

To further understand these dynamics, we plot both the surrogate loss $\widehat{L}_{<t}$ and the corresponding true loss $L_{<t}$ over epochs (Fig.3, center). While higher $\alpha$ values enable SGD to reduce $\widehat{L}_{<t}$ more effectively, this comes at the cost of significantly increased $L_{<t}$, indicating overfitting to the surrogate and degradation of the shared feature extractor $F$. This behavior arises because SGD cannot update $F$ effectively without gradients from past data. ES, being gradient-free, avoids this limitation and achieves a much lower $\widehat{L}_{<t}$ and correspondingly lower $L_{<t}$. As a result, ES outperforms SGD substantially, achieving an average accuracy improvement of approximately 32 percentage points, as illustrated in the right plot of Fig.3.

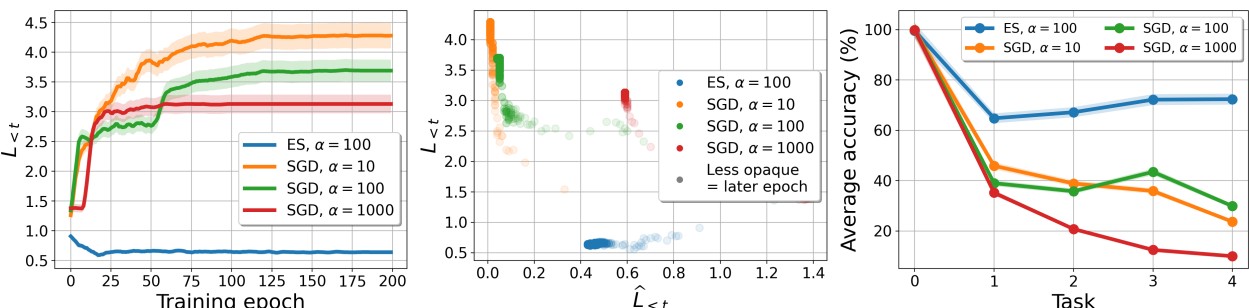

Figure 3: Comparison of ES and SGD optimizer for different $\alpha$ values on Fashion MNIST dataset. SGD is unable to minimize the $L_{<t}$ (left) although it minimizes the $\widehat{L}_{<t}$ better than ES (middle). This leads to worse average accuracy after each task (right).

**Does minimizing $\widehat{L}_{<t}$ reduce $L_{<t}$?** Our method relies on the key assumption that the surrogate loss $\widehat{L}_{<t}$, computed using the adapter network on transformed features, is correlated with the true loss $L_{<t}$ evaluated on the original data. To empirically validate this assumption, we track and plot the values of $\widehat{L}_{<t}$ and $L_{<t}$ across training epochs for each task, as shown in Fig. 4 (left and middle). The results on CIFAR100 and Aircraft datasets reveal a strong correlation between the two loss signals, indicating that minimizing $\widehat{L}_{<t}$ indeed leads to a corresponding decrease in $L_{<t}$.

To quantify this relationship, we compute the Pearson correlation coefficient between $\widehat{L}_{<t}$ and $L_{<t}$ over time. The resulting values of 0.953 (CIFAR100) and 0.995 (Aircraft), displayed in the upper-left corners of the respective plots, confirm a strong linear correlation between the surrogate and ground-truth losses. These findings support the validity of our approximation and justify the use of $\widehat{L}_{<t}$ as a reliable optimization objective in the absence of past-task gradients.

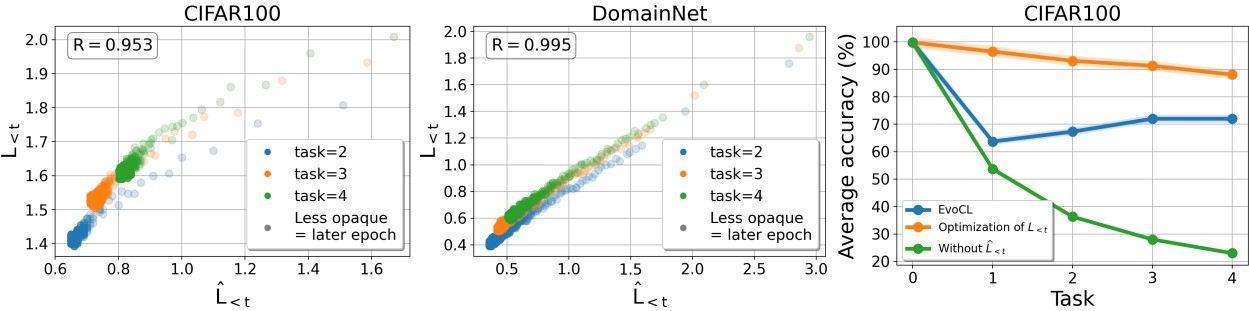

Figure 4: Correlation of approximated loss for previous tasks $\widehat{L}_{<t}$ and ground truth $L_{<t}$ (left, middle) after second, third and fourth task. Each point presents the loss values after a training epoch. We can see that losses are correlated and minimizing $\widehat{L}_{<t}$ decreases $L_{<t}$. EvoCL performs poorly without $\widehat{L}_{<t}$ in the loss function (right), and much better with $L_{<t}$ which is the upper-bound assuming we have access to old data.

**Influence of number of parameters.** We investigate how the number of trainable parameters affects continual learning performance across three methods: EvoCL, LwF, and FeCAM. As shown in the left plot of Fig. 5, EvoCL consistently achieves higher average incremental accuracy than both LwF and FeCAM across all parameter scales. Notably, EvoCL reaches strong performance (above 70%) even with as few as 5K parameters and maintains robust accuracy as the model size increases, exhibiting diminishing returns beyond 76K parameters. In contrast, LwF and FeCAM show a steeper dependency on model size, with substantial accuracy improvements only at larger scales (e.g., 76K+ parameters). The right plot further highlights the trade-off between forgetting and plasticity. EvoCL maintains a favorable balance, achieving lower forgetting at comparable or higher levels of plasticity compared to LwF and FeCAM, especially in the low-parameter regime (e.g., 1.4K–19K). LwF exhibits high plasticity but suffers from increased forgetting as parameters grow, while FeCAM displays moderate forgetting with rising plasticity. Overall, EvoCL demonstrates superior parameter efficiency, making it particularly suitable for memory-constrained continual learning scenarios.

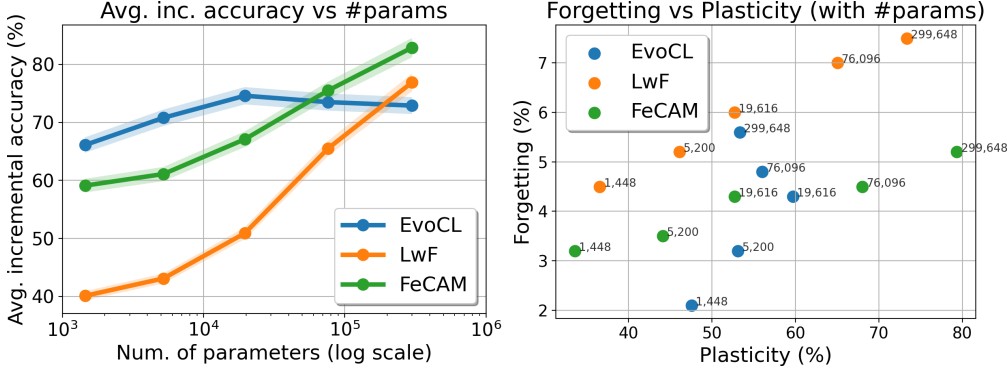

Figure 5: Comparison of different CL methods on Fashion MNIST for different number of utilized parameters. We see that EvoCL excels in low number of parameters.

**Ablation study.** We further assess the effectiveness of EvoCL through a detailed ablation study on the DomainNet benchmark split into 12 sequential tasks (Table 3). The full EvoCL configuration achieves a strong balance between final task performance ($A_{last} = 63.0\%$), forgetting (13.5%), and plasticity (72.5%) - while maintaining low memory usage (11.2MB). Removing the loss approximation component ($\widehat{L}<t$) causes a catastrophic collapse in performance ($A_{last} = 6.9\%$), along with extreme forgetting (84.7%), despite high plasticity (84.2%). This highlights the essential role of approximating past-task loss in the absence of exemplars. In contrast, removing the current task classification loss ($L_t$) yields surprisingly low forgetting (4.1%) but at the cost of zero plasticity and reduced accuracy, confirming that continual adaptation to new tasks is impossible without learning current task representations. Excluding the recursive transformation of features leads to a severe performance drop ($A_{last} = 34.6\%$), emphasizing the need to align stored features

with the current task's embedding space. Using SGD instead of evolutionary strategy significantly reduces both accuracy (42.8%) and plasticity, demonstrating the suitability of ES for optimizing non-differentiable objectives in EvoCL. Omitting the adapter's reconstruction loss ($L_{\mathrm{MSE}}$) also harms performance (31.6%), indicating that faithful transformation of stored features is critical for effective loss approximation. Finally, replacing latent features with raw exemplars leads to the highest accuracy (74.2%) and lowest forgetting (6.3%), but at a memory cost of over 60x, highlighting the trade-off EvoCL addresses under exemplar-free constraints.

Table 3: Ablation study on EvoCL. We report average accuracy (%), forgetting (%), plasticity (%) and memory usage on DomainNet split to 12 tasks. Full EvoCL includes all the components and performs optimally.

| **Variant** | $A_{last}$ ↑ | Forgetting ↓ | Plasticity ↑ | Memory (MB) |
|---|---|---|---|---|
| Full EvoCL | **63.0**±0.4 | 13.5±0.2 | 72.5±0.4 | 11.2 |
| w/o $\widehat{L}_{<t}$ | 6.9±0.1 | 84.7±0.5 | **84.2**±0.3 | 11.2 |
| w/o $L_t$ | 56.2±0.3 | **4.1**±0.1 | 0.0±0.0 | 11.2 |
| w/o Recursive $\psi$ | 34.6±0.3 | 60.9±0.3 | 71.4±0.3 | 11.2 |
| w/ SGD (no ES) | 42.8±0.2 | 37±0.3 | 67.3±0.2 | 11.2 |
| w/o $L_{\mathrm{MSE}}$ | 31.6±0.2 | 53.0±0.3 | 79.3±0.5 | 11.2 |
| w/ exemplars | 74.2±0.3 | 6.3±0.1 | 69.2±0.3 | 689.1 |

**Time complexity of EvoCL** To assess computational overhead, we compare the time complexity of EvoCL with representative EFCIL baselines on CIFAR100 split into 10 tasks (Table 4). Inference times are similar across methods, with EvoCL requiring 27.2 seconds - on par with LwF (27.1 s), EWC (27.2 s), and others. However, EvoCL exhibits significantly higher training time at 984.8 minutes, compared to FeTrIL (3.3 min), FeCAM (3.9 min), and DS-AL (3.7 min). This overhead stems from the use of ES optimizer that requires multiple models to be evaluated simultaneously and more epochs to converge. Despite this, EvoCL's performance in exemplar-free continual learning underscores a clear trade-off between training cost and performance.

Table 4: Time complexity of EvoCL and EFCIL baselines measured on CIFAR100 split into 10 tasks.

| | LwF | FeTrIL | FeCAM | EWC | DS-AL | EvoCL |
|---|---|---|---|---|---|---|
| Inference (sec) | 27.1±1.8 | 27.2±1.9 | 31.2±3.2 | 27.2±1.7 | 27.6±2.2 | 27.2±2.0 |
| Training (min) | 18.8±2.1 | 3.3±0.2 | 3.9±0.4 | 20.7±2.4 | 3.7±2.2 | 984.8±24.2 |

**Hyperparameter Sensitivity Analysis** To verify the algorithmic robustness of EvoCL under varying configurations, we perform a sensitivity analysis on three critical hyperparameters: the surrogate loss alignment scale ($\alpha$) from Eq. 2, the parent population size ($\mu$), and the offspring population size ($\lambda$). All evaluations are conducted using the DomainNet benchmark ($T = 12$ tasks). We report the final average accuracy ($A_{last}$) across all tasks along with the average forgetting. The unified empirical results are compiled in Table 5.

*Impact of Surrogate Alignment Scaling ($\alpha$).* The scaling parameter $\alpha$ regulates the strictness of the alignment between the latent feature buffer and the adapter space. When scaled insufficiently ($\alpha = 1$), the historical constraints are too loose to prevent catastrophic forgetting, which degrades overall performance (59.3%). Conversely, an overly strict penalty ($\alpha = 1000$) suppresses the model's plasticity, yielding the lowest forgetting rate (5.2%) but severely limiting the capacity to assimilate incoming tasks (54.1%). Our default setting ($\alpha = 100$) achieves the ideal balance, yielding optimal overall accuracy.

*Impact of Offspring Population Size ($\lambda$).* The parameter $\lambda$ controls the number of mutated candidate solutions evaluated per generation. As detailed in the empirical results, a restricted offspring budget ($\lambda = 32$) limits search coverage, causing a notable drop in performance (54.4%). Elevating $\lambda$ to our default configuration of 128 stabilizes the optimization significantly (63.0%). While continuing to scale the offspring size up to $\lambda = 2048$ yields our highest absolute accuracy (63.8%) and a minimal forgetting rate (7.5%), performance gains plateau heavily beyond $\lambda = 128$. Thus, the default configuration represents the most practical trade-off between task performance and computational overhead.

*Impact of Parent Population Size ($\mu$).* The parameter $\mu$ defines the number of top-performing individuals selected to propagate modifications into the subsequent generation. Restricting the parent selection pool too tightly ($\mu = 4$) under-samples effective parameter configurations, dropping the final average accuracy to 57.9%. Conversely, expanding the parent pool excessively ($\mu = 64$ and $\mu = 256$) dilutes the evolutionary selection pressure by allowing lower-performing candidates to influence the mutation trajectory, which increases the forgetting rate up to 9.4%. The default setting of $\mu = 16$ establishes the peak equilibrium required for efficient exploration and stable convergence.

Table 5: Unified hyperparameter sensitivity analysis of EvoCL on DomainNet ($T = 12$ tasks) under strict exemplar-free constraints. Bold rows indicate the default optimal configuration used across all primary experiments.

| Parameter | Value Explored | Average Accuracy ($A_{last} \uparrow$) | Forgetting Rate ($F_T \downarrow$) |
|---|---|---|---|
| Surrogate Alignment Scaling ($\alpha$) | $\alpha = 1$ | $59.3\% \pm 0.4$ | $18.4\% \pm 0.6$ |
| | $\alpha = 10$ | $61.8\% \pm 0.3$ | $12.1\% \pm 0.4$ |
| | $\alpha = 100$ **(Default)** | **63.0% $\pm$ 0.4** | **7.8% $\pm$ 0.3** |
| | $\alpha = 1000$ | $54.1\% \pm 0.5$ | $5.2\% \pm 0.2$ |
| Off-spring population size ($\lambda$) | $\lambda = 32$ | $54.4\% \pm 0.4$ | $14.9\% \pm 0.5$ |
| | $\lambda = 128$ **(Default)** | **63.0% $\pm$ 0.4** | **7.8% $\pm$ 0.3** |
| | $\lambda = 512$ | $63.1\% \pm 0.5$ | $7.8\% \pm 0.4$ |
| | $\lambda = 2048$ | $63.8\% \pm 0.4$ | $7.5\% \pm 0.5$ |
| Parent population size ($\mu$) | $\mu = 4$ | $57.9\% \pm 0.5$ | $10.2\% \pm 0.4$ |
| | $\mu = 16$ **(Default)** | **63.0% $\pm$ 0.4** | **7.8% $\pm$ 0.3** |
| | $\mu = 64$ | $62.2\% \pm 0.4$ | $8.6\% \pm 0.4$ |
| | $\mu = 256$ | $61.4\% \pm 0.7$ | $9.4\% \pm 0.4$ |

**Scope and limitations** While EvoCL establishes a robust proof-of-concept for gradient-free optimization in continual learning and outperforms strong EFCIL baselines, its current scope is verified on small to medium-scale benchmarks and fine-grained sub-networks. Large-scale experimental evaluations on full datasets like ImageNet-1k from scratch remain unavailable. This is primarily because population-based evolution strategies face increased computational and mutation complexity when navigating exceptionally large parameter dimensions. Future work will investigate localized or block-wise ES updates to scale this methodology to larger architectures and datasets.

## 5    Conclusions

In this work we introduced EvoCL, a gradient-free optimization approach for CL that mitigates catastrophic forgetting by approximating past task losses using an auxiliary adapter network. Rather than relying on gradient signals from previously seen data - which are unavailable in EFCIL scenario - EvoCL stores latent representations of class features and uses evolution strategy to optimize network parameters. This allows the method to maintain performance across sequential tasks without backpropagating through old data. A key contribution of EvoCL is its ability to jointly train both the feature extractor and the feature adaptation network, enabling the model to adapt to new tasks while retaining functional representations of earlier ones.

Despite its strengths, EvoCL has several limitations. First, due to the use of a lightweight architecture and limited parameter budget, the method may struggle to scale to larger or more complex datasets where higher model capacity is required. Second, the presented ES optimizer, while effective in this context, tends to be computationally expensive and sample-inefficient compared to gradient-based alternatives, which may impact training time and resource demands. Third, the success of EvoCL relies on the quality of the approximated loss function; inaccuracies in the auxiliary network's estimation of past task losses can degrade performance and lead to suboptimal updates.

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
