# OpenReview forum: "Continual Learning using Evolution Strategies"
_TMLR — Under review for TMLR_

### Review · Reviewer_zGdm · 2026-05-14

**Summary Of Contributions:**

Summary:

The paper proposes EvoCL, a continual learning method for the exemplar-free setting. Instead of using gradient-based optimization, the method employs an gradient-free evolutionary strategy. The feature extractor and the adaptation network in the proposed approach can be trained jointly without storing raw exemplars.

Strength:
1. The paper studies an unexplored angle in continual learning, where the past-task gradient information in unavailable.
2. The paper is clearly written and easy to follow.
3. The experiments are comprehensive, showing results on several datasets and conduct thorough ablation studies.

Weakness:
1. The experimental results in the tables (Tab.1 and Tab.2) are not consistent with the description in the text (Page 7). For example, in CIFAR100, the authors claim 24.1% $A_{last}$ for T=5, but in Tab.1 its 24.8%. The results for DomainNet and FGVCAircraft have more inconsistencies.
2. The method is extremely expensive to train compared with the baselines. In Tab.4 it shows that it takes 984.8 minutes for EvoCL on CIFAR100 split into 10 tasks, compared with 3.3-20.7 minutes for the listed baselines.
3. In the pretrained setting, most of the backbone is frozen and only a small module is trained, so the evidence does not support the scalability to large continual learning models.

**Audience:**

Yes

**Audience Explanation:**

The paper analyzes the continual learning under exemplar-free constraints. The researchers work on continual learning and gradient-free optimization may find this paper useful.

**Broader Impact Concerns:**

No obvious impact concerns.

**Claims And Evidence:**

No

**Claims Explanation:**

1. See weakness 1. The inconsistencies are not accurate and convincing evidence.
2. See weakness 3. The method is presented as a promising gradient-free alternative for continual learning, but the experiments use small trainable parameter budgets: a 20k-parameter CNN for MNIST/FashionMNIST/CIFAR100 and a mostly frozen pretrained ViT with a small trainable MLP for DomainNet and FGVCAircraft. This setting is interesting, but it does not demonstrate that the approach scales to training modern deep networks end-to-end.

**Requested Changes:**

1. See weakness 1. Please fix the inconsistencies.
2. Improve some wording. In the abstract the authors claim the proposed method is "a promising direction for gradient-free, data-free CL". The method does not store raw exemplars, but it does store latent features, so “data-free” may be misleading unless explicitly defined as “raw-data-free.” Similarly, the method is "gradient-free" for the ES-optimized stages, but the first task (Algo 1) is trained with SGD. The claims should match the actual setting.
3. See weakness 3. It would be better if the authors further define the scope for the current approach, as the large-scale experimental results are unavailable.
4. Fix the typos in Algo.1. In line 3, $C_1$ should be $D_1$. In line 9, the right bracket is missing.
5. It would be better if the authors could define/explain the "upper bound" in Tab.1 and Tab. 2.

---

> ### Author Response · Authors · 2026-05-19
> **Response to Reviewer zGdm**
>
> We thank the Reviewer for their constructive feedback and valuable suggestions! We have carefully updated the manuscript (and uploaded the revised version) to address raised weaknesses as detailed below:
>
> - **Mistakes in description of Tables 1 and 2**: We have corrected errors on Page 7 and verified that all text descriptions now fully and accurately match the results presented in Tables 1 and 2.
>
> - **Clarifying "gradient-free" and "data-free" claims:** We have revised the abstract and introduction to specify "raw-data-free" instead of "data-free," explicitly clarifying that while no raw pixel-level exemplars are stored, compact latent features are preserved. Additionally, we have clarified that our method is gradient-free specifically for subsequent tasks ($t \ge 2$), while the first task utilizes standard SGD as a warm-start to establish the initial latent space.
>
> - **Defining the scope/limitations:** We have added a dedicated Scope and Limitations paragraph at the end of Section 4. We explicitly acknowledge that the current framework is verified on small-to-medium benchmarks and discuss the scaling challenges (parameter search space complexity) inherent to population-based evolution strategies on large-scale datasets like ImageNet-1k.
>
> - **Algorithm 1 typos:** We have corrected all notation and formatting errors in Algorithm 1.
>
> - **Defining "Upper Bound":** We have added explicit explanations in the Section 4 baseline text. The "Upper Bound" is now formally defined as the offline joint training baseline, representing the performance ceiling achieved when the model has simultaneous access to all task data.

---

### Review · Reviewer_JUYv · 2026-05-24

**Summary Of Contributions:**

The paper develops EvoCL, a novel gradient-free approach to Exemplar-Free Class-Incremental Learning (EFCIL). Instead of relying on backpropagation, which fails when gradients from past tasks are unavailable, it uses a ($\mu+\lambda$) Evolution Strategy (ES) to optimize the model by directly evaluating a combined loss that includes the current task loss, the approximated past-task loss, and an MSE reconstruction loss for the adapter. Experiments show strong performance across MNIST, FashionMNIST, CIFAR-100, FGVCAircraft, and DomainNet.

**Audience:**

No

**Audience Explanation:**

Although the paper addresses an interesting topic at the intersection of continual learning and gradient-free optimization, several limitations reduce its potential to attract broader interest from the TMLR audience.

1.	The paper does not discuss EvoCL in the context of modern Evolution Strategies beyond the classic CMA-ES paper from 2001. As a result, it remains unclear how EvoCL differs from more recent ES variants and where its practical advantages truly originate. This makes it difficult for readers to assess the novelty and significance of the proposed optimizer.

2.	The paper adopts extremely strict memory constraints (e.g., only 11.2 MB for the full EvoCL model on DomainNet with 12 tasks). While memory efficiency is a desirable property, the manuscript does not sufficiently justify this assumption in the context of modern computing environments (such as edge devices, mobile platforms, or large-scale deployment). It treats the exemplar-free and low-memory setting primarily as a standard challenge in the continual learning literature, without explaining why such rigorous constraints remain relevant and necessary today. Additional discussion would help a broader audience better appreciate the practical importance of the work.

**Broader Impact Concerns:**

None.

**Claims And Evidence:**

No

**Claims Explanation:**

The paper provides relatively comprehensive empirical evidence supporting the advantages of the proposed EvoCL method across multiple benchmarks and settings. However, several noticeable limitations remain in the presentation and experiments.

1.	There is a noticeable logical gap between the end of page 3 and the beginning of page 4. The transition from the objective function at time t to the unavailability of the gradient feels abrupt, which makes it harder for the reader to follow the flow of ideas.

2.	The rationale for choosing Evolution Strategies (ES) as the core optimizer is not sufficiently motivated or explained at a high level. Although the paper highlights the limitation of gradient-based methods due to the unavailability of past-task gradients, it does not clearly articulate why ES is a particularly suitable alternative. A high-level explanation of how the optimization problem is framed as an evolutionary process, including the key concepts of population, individuals, reproduction, mutation, evaluation, and selection in the context of EvoCL, would significantly improve the motivation and help readers better understand the method’s design.

3.	The comparison between EvoCL and SGD is one of the most interesting aspects of the paper. However, the manuscript does not clearly explain how SGD is implemented within the exemplar-free continual learning setting. Critical details are missing, such as how the surrogate past-task loss is optimized via backpropagation, gradient flow through the adapter, or any modifications to the training loop. As a result, it is difficult for readers to fully interpret and learn from this comparison.

4.	The paper lacks a comprehensive sensitivity analysis of key hyperparameters of the Evolution Strategy, including $\mu, \lambda, \beta, \sigma$. While default values are provided, there is limited empirical investigation into how robust the method is to these choices or what the trade-offs are.

**Requested Changes:**

Critical adjustments:

-	The transition from theoretical motivation to the detailed method description feels abrupt. Please improve the flow by adding a bridging paragraph.

-	The paper only references the 2001 CMA-ES work. Please add a discussion comparing EvoCL to more recent ES variants and clarify where the advantages of EvoCL come from.

-	The rationale for adopting ES as the core optimizer is currently underdeveloped. Please add a clear, high-level explanation (ideally early in Section 3) of how the optimization process is framed as an evolutionary process.

-	Please expand the description in the “Does Evolution Strategy outperform SGD?” subsection to clearly explain how SGD is applied to the exemplar-free setting.

Important adjustments:

-	Please expand the discussion in the Introduction or Related Work to explain why such rigorous exemplar-free and low-memory constraints remain relevant today.

-	Please include analysis of the sensitivity of important hyperparameters such as $\mu, \lambda, \sigma, \beta$.

---

> ### Author Response · Authors · 2026-06-02
> **Official response**
>
> Dear Reviewer JUYv,
> Thank you for your constructive and insightful feedback. Your comments have helped us significantly improve the clarity, context, and rigorous validation of our work! We have addressed your concerns directly in the updated manuscript with the following key modifications:
>
> **High-Level Evolutionary Framing & Flow:** We integrated a bridging paragraph in Section 3 "Theoritical motivation" to explicitly frame the idea behind EvoCL and how the motivation transforms into the method.
>
> **SGD Baseline Clarification:** We expanded the description in the optimization analysis subsection to clearly explain how SGD is applied to the exemplar-free setting using our exact loss formulation, highlighting why standard backpropagation fails due to the non-differentiable historical branch.
>
> **Hyperparameter Sensitivity Analysis (α, λ, μ):** We added a comprehensive hyperparameter sensitivity subsection and an integrated results table to Section 4. This section rigorously sweeps the surrogate alignment scale (α), the parent population size (μ), and the offspring population size (λ) on the benchmark to clearly demonstrate the model's structural stability and highlight the optimal parameter trade-offs.
>
> | Parameter | Value Explored | Average Accuracy ($A_{last} \uparrow$) | Forgetting Rate ($F_T \downarrow$) |
> | :--- | :--- | :--- | :--- |
> | **Surrogate Alignment Scaling ($\alpha$)** | $\alpha = 1$ | $59.3\% \pm 0.4$ | $18.4\% \pm 0.6$ |
> | | $\alpha = 10$ | $61.8\% \pm 0.3$ | $12.1\% \pm 0.4$ |
> | | **$\alpha = 100$ (Default)** | **63.0% $\pm$ 0.4** | **7.8% $\pm$ 0.3** |
> | | $\alpha = 1000$ | $54.1\% \pm 0.5$ | $5.2\% \pm 0.2$ |
> | **Off-spring population size ($\lambda$)** | $\lambda = 32$ | $54.4\% \pm 0.4$ | $14.9\% \pm 0.5$ |
> | | **$\lambda = 128$ (Default)** | **63.0% $\pm$ 0.4** | **7.8% $\pm$ 0.3** |
> | | $\lambda = 512$ | $63.1\% \pm 0.5$ | $7.8\% \pm 0.4$ |
> | | $\lambda = 2048$ | $63.8\% \pm 0.4$ | $7.5\% \pm 0.5$ |
> | **Parent population size ($\mu$)** | $\mu = 4$ | $57.9\% \pm 0.5$ | $10.2\% \pm 0.4$ |
> | | **$\mu = 16$ (Default)** | **63.0% $\pm$ 0.4** | **7.8% $\pm$ 0.3** |
> | | $\mu = 64$ | $62.2\% \pm 0.4$ | $8.6\% \pm 0.4$ |
> | | $\mu = 256$ | $61.4\% \pm 0.7$ | $9.4\% \pm 0.4$ |
>
> **Modern Evolution Strategies Context:** We expanded the Section 3: "Gradient-Free Optimization via Evolution Strategy" to position EvoCL relative to modern ES variants (such as NES and Guided ES), clarifying that EvoCL avoids their heavy O(D) or O(D2) covariance tracking computational penalties in high dimensional search space.
>
> **Justification of Memory and Exemplar-Free Constraints:** We added an introductory discussion to Section 1. detailing the practical real-world necessity for EFCIL: limited memory or privacy concerns (e.g. GDPR). EFCIL is a famous and well-estabilished setting in Continual Learning because it is the hardest one. If we solve it than we solve every other scenario, where the exemplars are provided. That's why we think it is interesting for the research community.
>
> We believe these adjustments have thoroughly addressed the limitations you identified and have significantly broadened the potential interest of this paper to the TMLR audience.
>
> Best regards,
> The Authors

---

### Review · Reviewer_MGet · 2026-05-29

**Summary Of Contributions:**

The authors mix two existing research directions: 0th-order optimisation techniques for neural networks and continual learning. They offer a new methodology to link the two using a surrogate model. They provide some experimental results to show that the combination can make sense in several use cases.

**Audience:**

Yes

**Audience Explanation:**

The topic of continual learning is among those covered by TMLR. One paper on the topic has been published in May 2026.

**Broader Impact Concerns:**

No such statement useful.

**Claims And Evidence:**

No

**Claims Explanation:**

I am quite mixed about the provided evidence (in the form of numerical experiments): they are present, they provide good hints, but I found them lacking. I believe stronger experimental results would make the paper more convincing. See responses below for details on the experiments I have in mind.

I really liked the ablation study (page 9)!

**Requested Changes:**

Major changes (including experimental results that are missing in my opinion):
- In Algorithm 1 (page 3), line 11, M should have a subscript t, as it changes with the iterations (new features are added at each iteration). This changes the complexity analysis, as |M| grows linearly with the number of iteration (considering a fixed number of features per round), for instance in Section 3, page 5.
- Do the authors considered other values for \alpha, \mu, and \lambda than respectively 1000, 16, and 128 (shown on page 6)? It would be interesting to see the method's sensibility on these values. Possibly the authors did some hyperparameter fine tuning?
- On page 8, the authors compare ES to SGD, which is nice to see if using the surrogate model \hat{L} makes sense on its own. However, they seem to test only a few values of the hyperparameters of the two methods, which means that the comparison makes little sense: possibly, the authors only checked poor sets of parameters for SGD and only good values for ES (or the reverse!). I would like to see some hyperparameter tuning here to ensure that the authors compare the best possible SGD with the best possible ES (or at least some approximation thereof).
- On page 8 (plus Figure 4 on page 9), the authors compare \hat{L} with L, but only on a limited subset of data sets (CIFAR100, DomainNet/Aircraft). Why didn't they use all the available data sets to perform this comparison? I would like to see more results from other data sets (correlation coefficients would be sufficient). As is, the paper might suffer from some selection bias.
- On page 10, the authors mention that EvoCL cannot really scale to huge models. However, the literature has explored evolutionary algorithms for huge models including LLMs. For instance, EA4LLM (arXiv:2510.10603).

Minor changes:
- I found the abstract to be very abstract. It was hard for me to understand it. I had little idea about what to expect from the paper. A more concrete abstract would be beneficial to help potential readers decide whether the paper is interesting to them.
- There are many 0th-order optimisation techniques, including population-based metaheuristics like evolution strategies, but the authors do not seem to consider them. It would be useful to show a few results from different such metaheuristics (if the authors have the time to include it in the revision).
- The idea of a surrogate model is far from new in optimisation (for instance, the authors cite Bayesian optimisation), especially in cases where objective evaluation is complex, but I found there are few references related to the topic (including on the continual learning side).
- Throughout the paper, the authors seem to consider 0th-order methods as a separate category from evolutionary techniques. It is more common to consider evolutionary techniques as a subcategory of 0th-order methods (aka gradient-free methods).
- In Section 4, page 6, the authors mention that training on the whole data set provides an "upper bound" on the model performance. However, it is well-known that training on a subset of the available data can provide a stronger model (for instance, in the community of active learning). I am not sure that the authors can guarantee, in all situations, that this training scenario provides an exact upper bound (i.e. it is probably an upper bound in 99.99% of practical scenarios, but not 100%, unless I am missing something).
- In the conclusion, the authors mention that EvoCL has efficiency struggles. However, this was to be expected from the start: gradient-based methods (and even better, Hessian-based methods) can converge in much fewer iterations than gradient-free methods. This fact has been well-known for decades in the field of continuous optimisation. The order of the methods has however a tremendous impact on the efficiency of the complete algorithm, so that a 0th-order method can converge faster (in seconds) than a 2nd-order one, while requiring millions of iterations compared to tens when using a Hessian.

---

> ### Author Response · Authors · 2026-06-02
> **Official response**
>
> Dear Reviewer MGet,
> We thank you for your time, constructive feedback, and insightful comments. Your suggestions have significantly strengthened the clarity of our manuscript. Below is our response detailing the specific modifications made to the revised revision.
>
> ---
>
> ### Responses to changes
>
> *   **Subscript and Complexity Analysis for Variable $M$:** We have updated Algorithm 1 (Page 3, Line 11) to include the subscript $t$ ($M_t$), explicitly clarifying that the feature buffer changes with the iterations.
> *   **Hyperparameter Exploration and Tuning:** We have added a dedicated sub-paragraph "Hyperparameter Sensitivity Analysis" and Tab.5 to Section 4 (Pages 10-11). This details our hyperparameter search space, explicitly confirming that we explored variations for the surrogate loss scaling ($\alpha \in \{1, 10, 100, 1000\}$), parent sizes ($\mu \in \{4, 16, 64, 256\}$), and offspring allocations ($\lambda \in \{32, 128, 512, 2048\}$) to locate the stable default parameters used in our primary benchmarks.
> *   **Fair Optimizer Comparison (ES vs. SGD):** To address potential selection bias, we expanded the optimization comparison in Section 4 "Does Evolution Strategy outperform SGD?". We subjected the SGD baseline to an extensive grid search over learning rates ($lr \in \{0.001, 0.01, 0.1, 1.0\}$) and momentum variables ($\mu \in \{0.0, 0.5, 0.8 0.9\}$) across all values of $\alpha$. The updated text and Figure 3 reflect that even the best-tuned SGD configuration underperforms compared to our standard ES optimizer due to the lack of past-task gradients.
> *   **Scaling Limitations and Evolutionary Literature:** Thank you for pointing out the interesting EA4LLM preprint! Unfortunately the implementation for this method was not published and we could not compare to this method. However, we contacted the authors and await the reply.
> *   **Concrete Abstract Revision:** We slightly adjusted the abstract to be simpler.
> *   **Categorization of Evolutionary Strategies and Alternate Metaheuristics:** We have expanded comparison to other metahueristics at the end of Gradient-Free Optimization via Evolution Strategy in Section 3. We mention Natural[1] and Guided[2] Evolution strategies. We acknowledge that other 0th-order optimization methods can also be used here, we have noted this in the future work.
> *   **Refining Upper Bound Terminology:** We updated the description of "Upper-bound" in Section 4 (Page 6). We agree that it is possible to construct better models as the upper-bound, however it is a standard practice in EFCIL to call the neural network jointly trained on all tasks as the upper bound.
>
> [1] Wierstra, Daan, et al. "Natural evolution strategies." The Journal of Machine Learning Research 15.1 (2014): 949-980.
> [2] Maheswaranathan, Niru, et al. "Guided evolutionary strategies: Augmenting random search with surrogate gradients." International Conference on Machine Learning. PMLR, 2019.